# Technological Reliability and Efficiency of Wastewater Treatment in Two Hybrid Constructed Wetlands in the Roztocze National Park (Poland)

**Agnieszka Micek, Krzysztof Jóźwiakowski *** **, Michał Marzec** **and Agnieszka Listosz**

Department of Environmental Engineering and Geodesy, University of Life Sciences in Lublin, Leszczyńskiego 7, 20-069 Lublin, Poland; agnieszka.micek@up.lublin.pl (A.M.); michal.marzec@up.lublin.pl (M.M.); agnieszka.listosz@up.lublin.pl (A.L.)
* Correspondence: krzysztof.jozwiakowski@up.lublin.pl

**Abstract:** The paper presents the results of a 3-year study on the technological reliability and the efficiency of typical domestic wastewater treatment in two hybrid constructed wetland systems (CWs) located in the area of the Roztocze National Park (Poland). The studied objects consist of an initial settling tank and a system of two beds of the vertical flow (VF) and horizontal flow (HF) (VF–HF) type with reed and willow. The wastewater flow rate in the constructed wetlands systems (CWs) was 0.4 and 1.0 $m^3$/d, respectively. During the study period (2017–2019) 20 series of analyses were performed and 60 wastewater samples were collected. Based on the obtained results the effects of pollutant removal and the technological reliability were determined, which were specified with the Weibull method. The average efficiency of biochemical oxygen demand ($BOD_5$) and chemical oxygen demand (COD) removal was 96–99%. Slightly lower effects were obtained in the case of total phosphorus (TP) removal (90–94%), as well as for total suspended solids (TSS) (80–87%) and total nitrogen (TN) (73–86%) removal. The analysed CWs were characterised by 100% technological reliability for BOD5 and COD, as well as a good reliability for TSS and TP (87–100%) but slightly lower for TN removal (35–89%). Hybrid CWs of VF–HF type should be recommended to use in protected areas for wastewater treatment and water resources quality protection.

**Keywords:** technological reliability; efficiency of domestic wastewater treatment; constructed wetland; hybrid system; national park

## 1. Introduction

National parks are institutions of great socio-educational importance, therefore, their activities should be linked to the process of education about sustainable development and environmental protection [1]. In such protected areas not only nature but also air, soil and water protection is necessary [2,3].

The development of tourism and the functioning of protected areas, including national parks, is based on the existence of museums, foresters' lodges, shelters or tourist trails with places to rest. Such objects require appropriate sanitary infrastructure, which allows their proper functioning. For example, in Poland, according to the Act of 16 April 2004 on Nature Conservation "it is prohibited to build or reconstruct construction or technical facilities in national parks and nature reserves, with the exception of facilities and equipment serving the purposes of a national park or a nature reserve" [4]. Therefore, in such areas, it is necessary to build water supply and wastewater treatment systems that should not interfere with the natural environment [2,3]. Properly implemented water and wastewater management has a significant impact on limiting the eutrophication process of surface waters and reduces the degradation of groundwater quality [5–8].

In protected areas, the construction of sewerage systems and collective wastewater treatment plants is usually impossible due to the large buildings dispersion. The only solution is to build septic tanks or local wastewater treatment systems, which enable the management of the wastewater in the place of its creation. The operation of septic tanks is associated with frequent wastewater removal by septic trucks, which contributes to air pollution within the protected area. Moreover, septic tanks are not always tight and may contribute to the degradation of groundwater quality. Therefore, the best solution for wastewater management is to build a local wastewater treatment system, which should ensure high removal effects. One such solution that can be used to protect water quality on the protected areas, is the use of constructed wetland wastewater treatment plants, which have been used for more than 60 years worldwide [9,10] and for over 30 years in Poland [11].

At the beginning, in the 20th century usually one-stage constructed wetlands systems (CWs) with vertical flow (VF) or horizontal flow (HF) were used [12–16]. However, since the beginning of the 21st century, hybrid CWs have been used more and more often because they provide much more efficient pollutant removal [11,15,17–28]. The advantages of CWs are—among others—the low cost of their construction and operation and low energy requirements [29–31]. Jóźwiakowski, et al. [32] after a multicriterial analysis showed that the application of hybrid CWs is consistent with the idea of sustainable development. These systems fulfill all the sustainability criteria, in particular the ecological criterion, as they ensure a high efficiency of wastewater treatment with relatively small energy demand. The processes of pollutant removal in CWs of VF and HF type have already been described among others by Brix [33], Vymazal [27] and Vymazal and Kröpfelová [34].

According to Vymazal [10], research on CWs at the beginning of the 21st century mainly included: creation of various technological hybrid systems in order to obtain optimal effects of contaminants removal, and most of all nitrogen; the search for appropriate materials with high phosphorus removal capacity; identification of bacteria participating in wastewater treatment processes; modelling of hydraulics and contaminants removal processes in various types of CWs. In recent years, more and more studies have focused on the reliability of contaminants removal in CWs [35–45]. However, most of the papers mentioned concern the reliability of wastewater treatment in one-stage CWs, and only a few studies demonstrate the reliability of hybrid CWs. Therefore, it was decided to take up this topic and conduct some research in this area.

The aim of this work is to present 3-year results of a study conducted on the technological reliability and the efficiency of domestic wastewater treatment in two hybrid CWs of VF–HF type with common reed (*Phragmites australis* (Cav.) Trin. ex Steud) and willow (*Salix viminalis* L.) located in the area of the Roztocze National Park (Poland). In the analysed objects, reed and willow were used because the previous 25 years of research performed in Poland have shown that these plants effectively support the processes of pollutants removal in CWs [21].

Currently, there are few studies on the application and functioning of CWs in national parks and protected areas. The results of the research presented in this paper may be of significant practical importance, which will allow the implementation of CWs in other national parks and protected areas worldwide.

## 2. Material and Methods

### 2.1. Characteristics of the Roztocze National Park

The Roztocze National Park (RPN) is located in south-eastern Poland in a temperate, transitional climate zone. Climatic conditions in this area are formed in the summer by masses of polar sea air, and in the winter by polar continental air. The area of the RPN is characterised by a high level of insolation, with an average of 1683 h of sunshine per year, which gives about 4.4 h per day. The average annual air temperature in the RPN is 7.4 °C. The coldest month of the year is January (with an average temperature of −2.4 °C) and the warmest is July (with an average temperature of 19.0 °C).

The vegetation period in the RPN is 209 days [46]. Annual precipitation over the 1998–2012 period on the territory of the park ranged from 602 mm to 889 mm, with an average of 734 mm [47].

Underground waters in the RPN are found in the Upper Cretaceous carbonate–silica rocks and in carbonate and sandy-carbonate tertiary rocks. These waters are crack-layer waters with very large resources. Only in underground river valleys does water occur in sandy and sandy-gravel quaternary sediments [48]. Both groundwater and surface water in the RPN is of very good quality, so it is essential to protect it from degradation. In order to protect the quality of surface water and groundwater, in the RNP in 2014, some measures were taken to build three hybrid CWs for domestic wastewater flowing away from foresters' lodges. These objects have different amounts of inflowing wastewater and bed surfaces, but they contain the same plants, i.e., reed and willow. The conception of construction of hybrid CWs in the RNP was described by Jóźwiakowski et al. [49] and then preliminary research results concerning the operation of two hybrid CWs in the RNP in the first year of operation were presented in another paper [50].

### 2.2. Characteristics of the Experimental Facilities

For the analysis in this paper two objects located in the area of the RNP in the following towns were selected: Zwierzyniec (object No. 1) and Florianka (object No. 2). The studied wastewater treatment plants consist of a 3-chamber initial settling tank with a pumping station and a system of two CW beds of the VF–HF type. In the first bed with vertical flow (VF-type), common reed (*Phragmites australis* (Cav.) Trin. ex Steud) was used. In the second bed with horizontal flow (HF-type) willow (*Salix viminalis* L.) was used (Figure 1). In both objects, thick sand with granulation of 1–2 mm was used to fill the beds of VF and HF type. To protect ground water from contamination, a 1-mm high-density polyethylene (HDPE) foil lining was used to seal the beds.

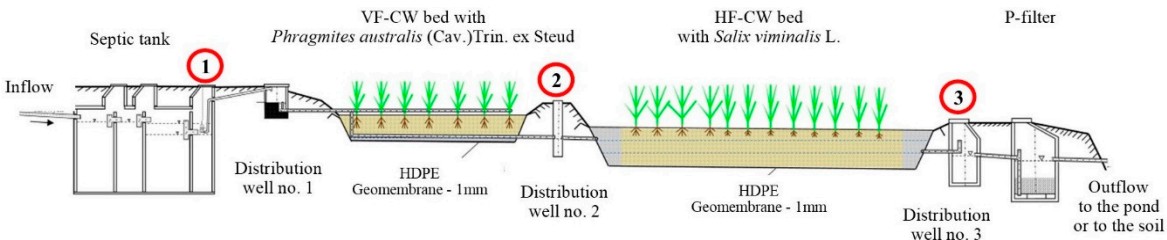

**Figure 1.** Longitudinal section of a hybrid CWs of VF–HF type in the Roztocze National Park [49]. Sampling points: 1, 2, 3. high-density polyethylene.

The final element of the wastewater treatment plants is the P-filters for phosphorus removal. However, this element was not the subject of research in this paper. Treated wastewater outflows into an infiltration pond (object No. 1) and to the soil (object No. 2) [49]. The wastewater flow rate during the whole research period in the tested systems was 0.4 and 1.0 $m^3$/day, respectively. The hydraulic load rate (HLR) in the investigated objects was 0.022 and 0.025 $m^3/m^2$/day, respectively, and the values were four times higher than in a one-stage HF-type CW (0.006 $m^3/m^2$/day) for which the technological reliability had been analyzed in earlier papers [37–39].

In Table 1, some technological parameters of the studied objects are presented and Figure 1 shows their longitudinal section.

### 2.3. Analytical Methods

Studies on the efficiency of pollution removal in two selected objects were conducted during the years 2017–2019. During this period, wastewater samples for analysis were taken in different seasons (spring, summer, autumn, winter) from: (1) the pumping station—after mechanical treatment in the initial settling tank, (2) after treatment in the first bed (VF type), (3) after treatment in the second bed (HF type). Tests of wastewater samples were carried out in the Laboratory of Water

and Wastewater Analytics of the Department of Environmental Engineering and Geodesy of the University of Life Sciences in Lublin (Poland). During the research period, 20 series of analyses were performed, during which, 60 wastewater samples from 3 points of each object were taken and analysed (Figure 1). The following parameters were determined: total suspended solids (TSS), BOD5, COD, total nitrogen (TN), total phosphorus (TP), pH, dissolved oxygen (DO), nitrate nitrogen, nitrite nitrogen and ammonium nitrogen.

**Table 1.** Technological parameters of the hybrid constructed wetlands systems (CWs) in the Roztocze National Park [49].

| Facility | No. 1 | No. 2 |
|---|---|---|
| Start of operation | 2014 | 2014 |
| Number of person equivalent | 4 | 10 |
| Amount of treated wastewater Q (m$^3$/day) | 0.4 | 1.0 |
| Active capacity of the initial settling tank V (m$^3$) | 4.9 | 4.9 |
| Area of bed of the CWs | | |
| Reed bed—VF (m$^2$) | I—18 | I—40 |
| Willow bed—HF (m$^2$) | II—30 | II—56 |
| Total area (m$^2$) | 48 | 96 |
| Bed area per 1 inhabitant (m$^2$/PE) | 12.0 | 9.6 |
| Bed depth (m) | VF—0.9 HF—1.2 | VF—0.9 HF—1.2 |
| Average hydraulic load of the first bed (m$^3$/m$^2$/day) | 0.022 | 0.025 |
| Hydraulic retention time in the bed (day) | VF—3.6 HF—24.0 | VF—3.2 HF—17.9 |
| Wastewater receiver | pond | soil |

Sampling, sample transportation, processing and analysis were completed according to relative Polish Standards of Wastewater Examination which are compatible with the American Public Health Association—APHA [51,52]. The laboratory equipment which was used to perform the analyses was described in an earlier paper [53].

The received measurement data were used to calculate: the mean, minimum, maximum concentration of pollutant values and standard deviation. The mean concentrations of the analysed pollutant parameters in the influent (Cin) and effluent from the VF and HF beds (Cout) were used to calculate the efficiency of pollution removal, which is given by formula 1:

$$\eta = 100 \times (1 - C_{out}/C_{in}) \ [\%] \tag{1}$$

Additionally, the efficiency of the tested hybrid CWs was analysed on the basis of mass removal rates (MRR) of the main pollutants contained in wastewater. MRR values were determined using formula 2.

$$MRR = \frac{C_{in} \cdot Q_{in} - C_{out} \cdot Q_{out}}{A} \ \left[ g/m^2/day \right] \tag{2}$$

where: A—surface area of the bed (m$^2$), Qin and Qout—average inflow and outflow of wastewater (m$^3$/day), Cin and Cout—average concentrations of pollutants in the wastewater inflowing and outflowing from the bed of VF or HF type (g/m$^3$). The calculated indicators are theoretical, because they are based on the assumption that the outflow of wastewater from particular elements of the treatment plant is equal to the inflow.

### 2.4. Statistical Analysis

The technological reliability of the wastewater treatment plants was determined with the Weibull method. The Weibull distribution is a general probability distribution function used for reliability testing and failure risk assessment over time. It has several uses, however, is often applied in device lifetime modelling and is flexible enough to replicate the key phases of the risk function run. With regard to wastewater treatment, the reliability analysis based on the Weibull distribution allows to determine the probability of occurrence of a certain value of pollution indicators in treated wastewater, and thus to assess the risk of exceeding the limit values. Thanks to this, it is possible to determine the time of defective operation of the treatment plant, which may be useful in making a decision regarding the operation of the treatment plant and its modernization [37–42]. The Weibull distribution is characterised by the following probability density function:

$$f(x) = \frac{c}{b} \cdot \frac{x - \theta}{b}^{(c-1)} \cdot e^{-\left(\frac{x-\theta}{b}\right)^c} \tag{3}$$

where: $x$—a variable describing the concentration of a pollution parameter in the treated effluent, $b$—scale parameter, $c$—shape parameter, $\theta$—position parameter.

Assuming: $\theta < x, b > 0, c > 0$.

The reliability analysis was carried out separately for 5 indicators (TSS, BOD5, COD, TN, TP), taking into account their values in treated wastewater discharged into the receiver ($n = 20$). The analysis consisted of the estimation of the Weibull distribution parameters using the maximum-likelihood method and the verification of the null hypothesis that the analysed variable could be described by the Weibull distribution. According to Dodson's recommendations [54] for the number of samples exceeding 15, the null hypothesis was verified with the Hollander–Proschan test at the significance level of 0.05.

The reliability was determined from the cumulative distribution function plotted in the graphs, taking into account the normative values of the indicators specified in the Regulation of the Minister of Maritime Economy and Inland Navigation [55] for wastewater discharged from treatment plants of up to 2000 PE (person equivalent): TSS—50 mg/L, BOD5—40 mg/L, COD—150 mg/L, TN—30 mg/L and TP—5 mg/L.

In Poland, total nitrogen and total phosphorus concentrations in treated wastewater are not taken into account when assessing the operation of small wastewater treatment plants (<2000 PE) in rural areas. However, because of the fact that the tested objects are located in the area of the national park, standard values applicable in Poland for wastewater discharged into lakes and their tributaries—as well as directly into artificial water reservoirs located in flowing waters—were adopted for the nitrogen and total phosphorus [55]. The analysis of the technological reliability of the studied objects was carried out using the Statistica 13 software.

## 3. Results and Discussion

### 3.1. The Efficiency of Pollutants Removal Processes

In the analysed systems, the concentrations of pollutants in wastewater flowing into the VF type beds were significantly lower than in raw wastewater flowing into the initial settling tanks (the first element of the plant), as described in an earlier study [53]. The average values of pollutants in wastewater inflowing to the 1st VF type beds in the studied systems are presented in Tables 2 and 3. The received values were similar to the ones presented in the literature for mechanically treated wastewater in the initial settling tanks [37,38,40–42].

**TSS.** Suspended solids that are not removed in pre-treatment system are effectively removed by processes of filtration and settlement which occur in the CWs [56]. In the studied objects No. 1 and No. 2, filtration and sedimentation processes were carried out in VF type beds with the highest efficiency, where the effects of total suspended solids removal were 58% and 73%, respectively (Figure 2), and MRR was 1.67 and 1.28 g/m$^2$/day, respectively (Table 4).

**Table 2.** Concentrations of pollutants in different steps of CWs in object No. 1.

| Parameters | 1—Inflow to the 1st Bed (VF) | | | | 2—Outflow from the 1st Bed (VF) | | | | 3—Outflow from the 2nd Bed (HF) | | | |
|---|---|---|---|---|---|---|---|---|---|---|---|---|
| | min | max | $\bar{x}$ | $\sigma$ | min | max | $\bar{x}$ | $\sigma$ | min | max | $\bar{x}$ | $\sigma$ |
| pH | 7.0 | 8.05 | - | 0.25 | 6.06 | 7.68 | - | 0.38 | 6.62 | 7.51 | - | 0.27 |
| Dissolved oxygen (mg/L) | 0.02 | 1.64 | **0.55** | 0.50 | 2.02 | 5.94 | **4.32** | 0.96 | 1.3 | 8.32 | **4.90** | 1.72 |
| Total suspended solids (mg/L) | 56.0 | 286 | **129** | 64.3 | 8.9 | 107 | **54.0** | 30.5 | 3.7 | 53.1 | **26.7** | 15.3 |
| BOD$_5$ (mg/L) | 193 | 345 | **275** | 38.6 | 3.4 | 149 | **15.4** | 31.8 | 1.1 | 11.2 | **3.5** | 2.4 |
| COD$_{Cr}$ (mg/L) | 575 | 1220 | **785** | 148 | 24.0 | 630 | **88.4** | 130 | 5.2 | 82.0 | **34.8** | 15.3 |
| Ammonium nitrogen (mg/L) | 78.0 | 139 | **110** | 18.9 | 0.28 | 58.0 | **17.0** | 16.5 | 0.12 | 41.4 | **11.4** | 10.8 |
| Nitrate nitrogen (mg/L) | 0.10 | 2.70 | **1.03** | 0.66 | 5.01 | 75.20 | **49.4** | 16.06 | 1.70 | 57.10 | **23.6** | 15.1 |
| Nitrite nitrogen (mg/L) | 0.01 | 0.41 | **0.21** | 0.10 | 0.06 | 14.95 | **1.93** | 3.18 | 0.04 | 1.14 | **0.41** | 0.27 |
| Total nitrogen (mg/L) | 82.9 | 209 | **144** | 29.4 | 55.0 | 130 | **81.4** | 20.3 | 9.1 | 95.2 | **39.9** | 20.8 |
| Total phosphorus (mg/L) | 14.2 | 71.8 | **27.2** | 13.7 | 9.1 | 27.0 | **12.7** | 4.4 | 0.1 | 7.0 | **2.6** | 2.0 |

**Table 3.** Concentrations of pollutants in different steps of CWs in object No. 2.

| Parameters | 1—Inflow to the 1st Bed (VF) | | | | 2—Outflow from the 1st Bed (VF) | | | | 3—Outflow from the 2nd Bed (HF) | | | |
|---|---|---|---|---|---|---|---|---|---|---|---|---|
| | min | max | $\bar{x}$ | $\sigma$ | min | max | $\bar{x}$ | $\sigma$ | min | max | $\bar{x}$ | $\sigma$ |
| pH | 6.88 | 7.48 | - | 0.15 | 6.35 | 7.46 | - | 0.26 | 7.52 | 8.35 | - | 0.26 |
| Dissolved oxygen (mg/L) | 0.02 | 2.33 | **0.71** | 0.64 | 2.84 | 5.90 | **4.59** | 0.67 | 2.35 | 10.7 | **7.24** | 2.01 |
| Total suspended solids (mg/L) | 25.0 | 135 | **70.6** | 26.9 | 4.9 | 46.8 | **19.5** | 9.2 | 2.5 | 25.5 | **9.9** | 6.9 |
| BOD$_5$ (mg/L) | 22.5 | 295 | **123** | 72.7 | 0.9 | 107 | **16.7** | 21.9 | 0.25 | 7.6 | **3.0** | 1.8 |
| COD$_{Cr}$ (mg/L) | 188 | 700 | **390** | 167 | 5.0 | 74.0 | **35.8** | 158 | 2.0 | 27.3 | **15.2** | 6.9 |
| Ammonium nitrogen (mg/L) | 59.0 | 95.0 | **77.5** | 9.2 | 0.03 | 63.1 | **6.6** | 13.9 | 0.01 | 6.2 | **0.49** | 1.4 |
| Nitrate nitrogen (mg/L) | 0.01 | 1.80 | **0.62** | 0.52 | 7.77 | 81.4 | **50.4** | 15.7 | 0.90 | 41.2 | **11.5** | 12.1 |
| Nitrite nitrogen (mg/L) | 0.01 | 0.27 | **0.10** | 0.05 | 0.07 | 1.61 | **0.64** | 0.39 | 0.01 | 3.04 | **0.25** | 0.68 |
| Total nitrogen (mg/L) | 62.0 | 146 | **98.6** | 19.8 | 46.0 | 103 | **71.3** | 14.0 | 2.0 | 40.8 | **13.8** | 13.1 |
| Total phosphorus (mg/L) | 10.2 | 48.3 | **19.4** | 9.9 | 2.89 | 22.2 | **9.2** | 5.8 | 0.2 | 2.5 | **1.1** | 0.5 |

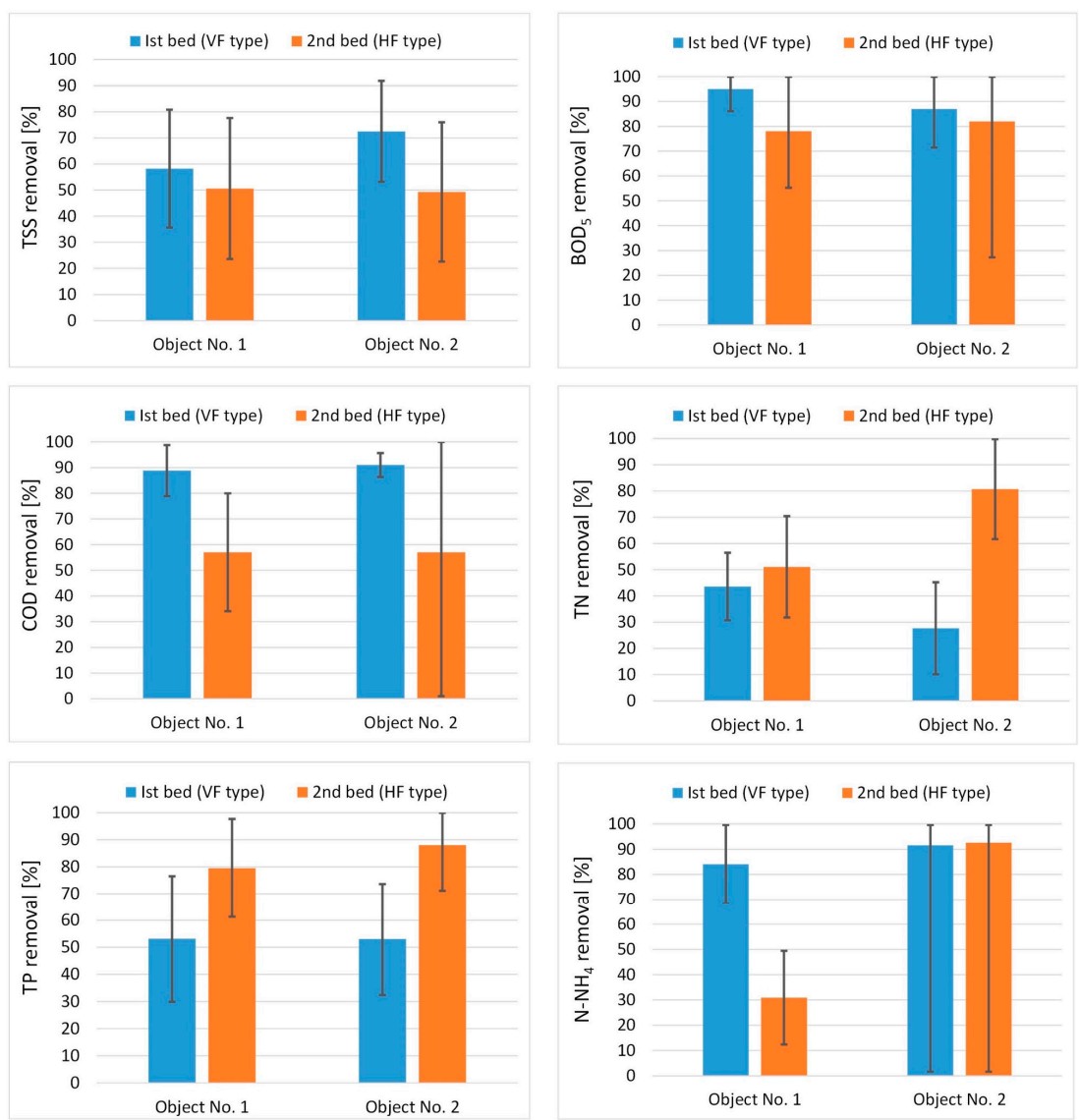

**Figure 2.** Average efficiency of pollutants removal in VF and HF beds of the studied CWs.

**Table 4.** Average pollutant load (APL) and mass removal rates (MRR) in the studied VF–HF CWs (in g/m$^2$/day).

| Object | | No. 1 | | | No. 2 | | |
|---|---|---|---|---|---|---|---|
| **Parameters** | | **VF** | **HF** | **VF–HF** | **VF** | **HF** | **VF–HF** |
| TSS | APL | 2.87 | 0.72 | 1.08 | 1.77 | 0.35 | 0.74 |
| | MRR | 1.67 | 0.36 | 0.85 | 1.28 | 0.17 | 0.63 |
| BOD$_5$ | APL | 6.11 | 0.21 | 2.29 | 3.08 | 0.30 | 1.28 |
| | MRR | 5.77 | 0.16 | 2.26 | 2.66 | 0.24 | 1.25 |
| COD | APL | 17.44 | 1.18 | 6.54 | 9.75 | 1.23 | 4.06 |
| | MRR | 15.48 | 0.71 | 6.25 | 8.03 | 0.96 | 3.90 |
| TN | APL | 3.20 | 1.09 | 1.20 | 2.47 | 1.27 | 1.03 |
| | MRR | 1.39 | 0.55 | 0.87 | 0.68 | 1.03 | 0.88 |
| N-NH$_4$ | APL | 2.44 | 0.23 | 0.92 | 1.94 | 0.12 | 0.81 |
| | MRR | 2.07 | 0.07 | 0.82 | 1.77 | 0.11 | 0.80 |
| TP | APL | 0.60 | 0.17 | 0.23 | 0.49 | 0.19 | 0.20 |
| | MRR | 0.32 | 0.13 | 0.21 | 0.22 | 0.17 | 0.19 |

Significantly higher effects of TSS removal (84–86%) were obtained in beds with vertical flow with common reed in two similar hybrid VF–HF type systems [20]. The mean effects of TSS removal in two parallel VF type beds with common reed and manna grass were 71% and 84%, respectively [38]. Quite high effects of total suspended solids removal (86.9%) were also obtained in the VF type bed with giant miscanthus, which was the first element of a hybrid VF–HF type system that was examined by Marzec et al. [41]. Past experience from various countries around the world indicates that TSS removal efficiency in VF type CWs with common reed ranges from 85–90% [9,13,57].

The efficiency of total suspended solids removal in the studied VF beds was lower than in similar objects of this type because the wastewater inflowing to the beds contained a small concentration of TSS (71–129 mg/L) (Tables 2 and 3, Figure 3), as a large part of them (42–60%) was removed in 3-chamber settling tanks [53].

It was shown that in the studied objects No. 1 and No. 2 the average effects of TSS removal in HF type beds were lower than in VF type beds and amounted to 51% and 49%, respectively (Figure 2), and MRR was also lower and amounted to 0.36 and 0.17 $g/m^2$/day, respectively (Table 4). Significantly lower effects of TSS removal (26–32%) were obtained in two beds with horizontal flow with willow in two similar hybrid VF–HF type objects [20]. Better TSS removal effects (54%) were obtained in a HF type bed with Jerusalem artichoke in a VF–HF hybrid system [42]. Slightly higher effects of TSS removal—62% were obtained by Marzec et al. [41] in a HF bed with Virginia mallow in a hybrid VF–HF system. Past experience from various countries around the world indicates that the TSS removal efficiency in HF type CWs with different plants ranged from 56% to 83% [9,15,26,58]. The diversified efficiency of TSS removal in different HF systems around the world is probably caused by the operation of these facilities in different climatic conditions and the use of different plants and materials to fill the beds, as well as the different depths of the beds, respectively (Figure 4). Slightly higher effects of TSS removal (88–89%) were previously obtained in two similar hybrid VF–HF objects with reed and willow in Poland [20] and in the VF–HF type objects (92–94%) studied by Marzec et al. [41,42]. Slightly lower effects of total suspended solids removal—78% on average—than those found in the analysed systems were observed in the VF–HF type object in Paistu, Estonia, which operates in a relatively cool climate (with average daily air temperatures ranging from −7.7 to 15.2 °C) [59]. However, a large number of hybrid CWs (VF–HF, or HF–VF type and some more developed objects) in the world, ensures the efficiency of total suspended solids removal at the level of 82–89% [11,23,60], i.e., similar to that obtained in the studied objects. Recently, there is an increasing number of such objects, e.g., in Nepal [61,62], South Korea [63], Spain in the Canary Islands [25], Ireland [64] and Japan [65], where the effects of total suspended solids removal are as high as 96–99%.

The study showed that the average content of total suspended solids in treated wastewater discharged from the examined facilities was 26 and 10 mg/L, respectively, and was significantly lower than the limit value (50 mg/L) specified in the Polish Regulation [55].

The average effects of TSS removal in both tested VF–HF hybrid systems were 80% and 87%, **BOD5 and COD**. It was shown that organic matter in CWs is decomposed by both aerobic and anaerobic microbial processes, as well as by sedimentation and filtration of particulate organic matter [34]. The intensities of aerobic and anaerobic degradation are highly dependent on the oxygen content in the wastewater flowing through the CWs [15]. Brix and Arias [66] indicated that very good aerobic conditions for organic matter decomposition are found in systems with vertical flow. Other studies [67,68] showed that the oxygen transport capacity in horizontal flow systems is insufficient for full aerobic pollutant decomposition and that anaerobic processes predominate there.

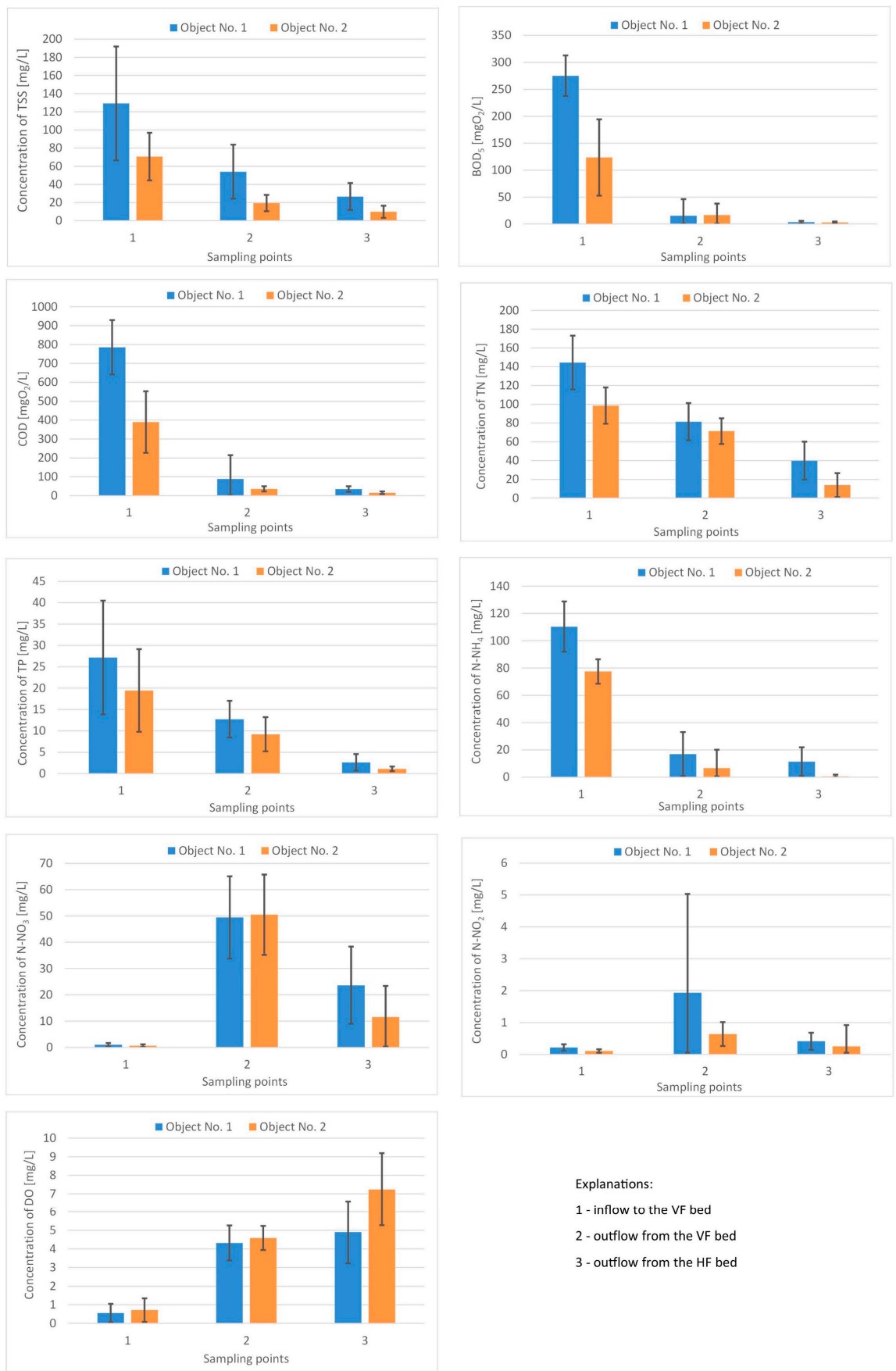

**Figure 3.** Average pollutant concentrations in different steps of wastewater treatment.

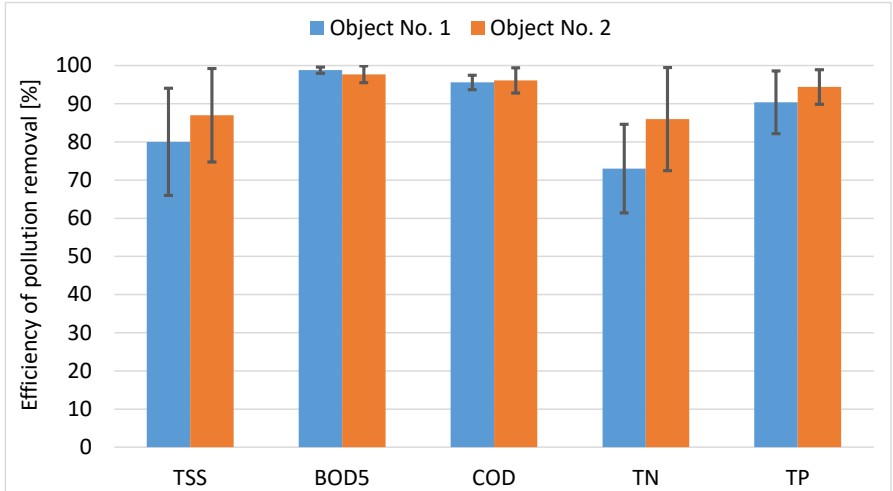

**Figure 4.** Average efficiency of pollutants removal in two hybrid VF–HF type CWs in the Roztocze National Park.

The conducted research showed that the VF type beds provided very good conditions for organic matter decomposition, as evidenced by the concentrations of dissolved oxygen in the wastewater outflowing from these beds that ranged from 4.32 to 4.59 mg/L (Figure 3). Slightly lower dissolved oxygen concentrations (2.4 to 3.7 mg/L) were observed by Gajewska and Obarska-Pempkowiak [69] in the outflow from the VF type beds in some objects in Wiklino and Wieszyno. On the other hand, Gikas et al. [70] found that in the outflow from the VF type system in Greece, the average $O_2$ concentration was 2.7 mg/L. The obtained research results indicate that the oxygen concentration in the wastewater outflowing from the analysed VF type beds was higher than previously reported in the literature [69,70], which proves that these beds provided better conditions for the decomposition of organic matter. Appropriate oxygen conditions resulted in the VF type beds having $BOD_5$ removal effects of 95% and 87%, respectively, and in the case of COD—89% and 91%, respectively (Figure 2). The MRR in vertical flow beds was 5.77 and 2.66 $g/m^2/day$ (for $BOD_5$) and 15.48 and 8.03 $g/m^2/day$ (for COD), respectively (Table 4). Similar effects of removal of $BOD_5$ (90–94%) and COD (86–92%) were obtained in two vertical flow beds with reed in two similar hybrid VF–HF type objects [20]. The mean effects of $BOD_5$ and COD removal in two parallel VF type beds with common reed and manna grass were slightly higher and amounted to 94–96% and 94%, respectively [41]. Quite high effects of $BOD_5$ and COD removal (97% and 95%, respectively) were also obtained in the VF type bed with giant miscanthus, which was the first element of the VF–HF hybrid system studied by Marzec et al. [42]. Past experience from various countries around the world indicate that the efficiency of BOD5 and COD removal in VF type CWs with common reed range from 87% to 96% and from 82% to 87%, respectively [9,13,57].

It was shown that in the studied objects No. 1 and No. 2, the effects of BOD5 and COD removal in HF type beds were lower than in VF type beds and amounted to 78% and 82% for BOD5 and 57% for COD (Figure 2). MRR was also significantly lower than in the VF type beds and amounted to 0.16 and 0.24 $g/m^2/day$ for BOD5 and 0.71 and 0.96 $g/m^2/day$ for COD (Table 4). Lower effects of BOD5 and COD removal (64% and 53%) were obtained by Marzec et al. [41] in HF type bed with Virginia mallow in a hybrid VF–HF treatment plant. Significantly lower effects of removal of BOD5 (36–42%) and COD (24–31%) were obtained in two horizontal flow beds with *Salix Viminalis* in two similar hybrid VF–HF plants [20]. On the other hand, Marzec et al. [42] studied the HF type bed with Jerusalem artichoke in the VF–HF hybrid system and obtained the efficiency of BOD5 and COD removal at the level of 88% and 50%, respectively. Past experience from various countries around the world indicates that the efficiency of BOD5 and COD removal in HF type CWs with different plants ranges from 75–85

and 67–75%, respectively [9,15,26,58]. The presented results from the world literature regarding the removal of BOD5 and COD in HF type beds, are similar to those obtained in the studied objects.

The average efficiency of BOD5 and COD removal in both tested hybrid VF–HF systems was very high and for BOD5 amounted to 99% and 98%, respectively, and 96% for COD (Figure 4). Similar removal efficiencies for BOD5 (97–99%) and COD (96–98%) were observed in HF–VF hybrid systems in Nepal [61], in VF–HF type systems in South Korea [63], as well as in VF–HF type objects studied in Poland [41,42]. Slightly lower removal effects in the case of BOD5 (94–96%) and COD (91–94%) were previously obtained not only in two similar hybrid VF–HF type objects with reed and willow in Poland [20], but also in a HF–VF type CWs located near a hotel in Florence, Italy [23]. Significantly lower effects of removal of BOD5 (86–91%) and COD (80–90%), than those observed in the analysed objects, were found in various configurations of hybrid systems: in the Canary Islands, Spain, in a Gran Canaria object [25], in Ireland in a Colecot object [64], in Poland in the Darżlubie, Wieszyno, Sarbsk and Schodno objects [11], in Nepal in a Thimi object [62] and in Japan on the island of Hokkaido in an Embestu object [65].

The study showed that the mean values of BOD5 and COD in treated wastewater discharged from the studied objects were 3–4 mg/L and 15–35 mg/L, respectively (Figure 3), and were many times lower than the limit values (40 and 150 mg/L) specified in the Polish Regulations [55].

**Total nitrogen and ammonium nitrogen.** The process of biological nitrogen removal in wastewater treatment plants proceeds with the participation of three groups of bacteria: ammonifying, nitrifying and denitrifying bacteria [71]. The presence of all three of these microbial groups in the particular elements of hybrid CWs is necessary for the efficient transformation of the organic nitrogen compounds present in wastewater into forms that are available for plants [14]. Each of these bacterial groups requires a different environment for optimal growth, and therefore, CWs should be designed in such a way as to ensure suitable conditions for the correct functioning of the individual nitrogen removal processes [72]. According to Benham and Mote [73], the processes of nitrification and denitrification are the primary mechanisms of nitrogen removal in CWs. The processes of oxidation, adsorption by the material filling that constitutes bed, and nitrogen uptake by plants play a much smaller role in nitrogen elimination [26]. According to Vymazal [27], in one-stage CWs it is impossible to achieve high nitrogen removal effects because they cannot provide both aerobic and anaerobic conditions at the same time. However, this is possible in hybrid systems with vertical and horizontal wastewater flow, in which nitrification and denitrification processes take place with various degrees of intensity. Vertical flow systems allow effective nitrification and $N-NH_4^+$ removal but denitrification is very limited. On the other hand, horizontal flow systems have favourable conditions for denitrification and are less able to oxidise ammonium nitrogen. Therefore, in order to achieve high nitrogen removal effects, hybrid CWs with different configurations are currently recommended [27].

In the studied objects—No. 1 and No. 2 in the VF type beds—not only a relatively high efficiency of ammonium nitrogen removal was found: 84% and 92%, respectively (Figure 3), but also a significant decrease in its concentration from 78–110 mg/L in the inflow to 7–17 mg/L in the outflow from these beds was noted (Figure 3). The MRR for VF beds was 2.07 and 1.77 $g/m^2$/day, respectively (Table 4). Simultaneously, an increase in the nitrate, nitrite and dissolved oxygen concentration was observed in the outflow from the VF beds (Figure 3). The obtained results prove that the VF type beds ensure an effective nitrification process and effective elimination of $N-NH_4^+$. Previous studies by Jóźwiakowski and Wielgosz [74], carried out in a hybrid VF–HF type CWs system with reed and willow, also showed that the VF type bed ensured an effective (91%) removal of ammonium nitrogen and efficient nitrification process. The authors showed that in wastewater flowing out of the first bed with common reed, the highest number of nitrifying bacteria of the I and II phase were recorded—64.6 and 25.1 NPL/mL, respectively, and increased concentrations of $N-NO_3^-$ (23.5 mg/L) and $N-NO_2^-$ (0.53 mg/L) were observed. These wastewater samples also contained a small content of $N-NH_4^+$—5.1 mg/L and the average dissolved oxygen concentration was 2.82 mg/L.

The research shows that the HF type beds in objects No. 1 and No. 2 provided further ammonium nitrogen removal. MRR for the HF beds was much smaller than for the VF beds and amounted to 0.07 and 0.11 g/m$^2$/day, respectively (Table 4). On the other hand, the effects of N-NH$_4^+$ removal were very varied and amounted to 31% and 93%, respectively (Figure 2). Additionally, the concentrations of ammonium nitrogen in the outflow from the HF type beds differed significantly (Figure 3). These differences could have been caused by aerobic conditions in the HF beds in objects No. 1 and No. 2, as evidenced by the concentrations of dissolved oxygen in the outflow from these beds, which were 4.90 and 7.24 mg/L, respectively. The studies show that the HF bed in object No. 2 provided good conditions for the course of both nitrification and denitrification processes, which is evidenced by the low concentrations of ammonium, nitrate and nitrite nitrogen in the wastewater flowing out of this bed, which are much lower than in the outflow from the HF bed in object No. 1 (Figure 3). On the other hand, in the HF bed in object No. 1, the denitrification process mainly took place. Haberl et al. [13] and Seo et al. [75] had previously stated that HF beds (with willow) can be successfully used in hybrid systems as a denitrification deposit for removing nitrates that remain after treatment in the nitrification VF type bed (with reed). Seo et al. [75], who studied the possibility of N-NO$_3^-$ removal in CWs using willow species such as *Salix gracilistyla*, *Salix chaenomeloides*, *Salix koreensis,* demonstrated a removal efficiency at the level of 59%. On the other hand, Jóźwiakowski and Wielgosz [74], in the HF type bed with *Salix viminalis* L., demonstrated a 55% level of efficiency for N-NO$_3^-$ removal. The study showed that the effects of nitrate nitrogen removal in the examined HF type beds in objects No. 1 and No. 2 were 52% and 77%, respectively.

The average effects of N-NH$_4^+$ removal in both tested VF–HF hybrid systems were 90% and 99%, respectively. A similar high efficiency of ammonium nitrogen removal (92%) was obtained in the HF–VF type object in Nepal [61]. Similar effects of ammonium nitrogen removal (88–91%) were also obtained earlier in two similar hybrid VF–HF type systems with reed and willow in Poland [20]. On the other hand, much smaller effects of N-NH$_4^+$ removal (84%) were obtained in VF–VF–HF type CWs in Ireland—84% [64], in a HF–VF system in Poland—79% [20] and in a VF–VF–HF system in the Czech Republic—78% [76]. In the HF–VF type object in Nepal the efficiency of ammonium nitrogen removal was only 69% [62].

The efficiency of total nitrogen removal in HF type beds (51–81%) was much higher than in the VF type beds (28–44%) (Figure 2). The mean efficiency of total nitrogen removal in both studied VF–HF type systems was 73% and 86%, respectively (Figure 4), and for total nitrogen MRR in the analysed systems it was 0.87–0.88 g/m$^2$/day (Table 4). Additionally, reasonably good total nitrogen removal effects (67–88%) were obtained in hybrid systems in Poland [14,41], South Korea [63,77], Spain [25], China [61] and Japan [65]. Lesser effects of total nitrogen removal (61–66%) were provided by similar VF–HF systems operating in Poland [20,42] and Estonia [59] and a HF–VF system in Denmark [17]. Even lower effects of total nitrogen removal (59–60%) were found in various hybrid systems in Italy [23] and Poland [11]. Research conducted by Jóźwiakowski [20] also shows that the VF–HF bed configuration is more effective in terms of total and ammoniacal nitrogen removal, than the HF–VF configuration. Such a trend was also previously observed in Norway [78], Austria [79], France [80], Ireland [64] and Germany [81]. Moreover, the study shows that the efficiency of total nitrogen removal is usually about 10–20% higher in hybrid CWs than in one-stage systems [20,38].

**Total phosphorus.** According to Cooper et al. [82] phosphorus can be removed from wastewater in CWs mainly through sorption and plant uptake. Lantzke et al. [83] stated that the only permanent mechanism for removing phosphorus in CWs is the felling and harvesting of plants from beds. However, Brix [84] considered that the amount of this element that can be removed by plant collection is small. According to Vymazal [15], only about 10% of phosphorus and nitrogen is removed in CWs with harvested plant biomass. It was shown that the efficiency of phosphorus removal in the initial period of CWs' exploitation is very high and then decreases after some time due to the loss of sorption capacity by the filter material [38,85].

In the studied objects No. 1 and No. 2, the process of total phosphorus removal was carried out in HF type beds with the highest efficiency, with the effects of its removal being 80% and 88%, respectively. Much smaller effects of total phosphorus removal were obtained in VF type beds, 53% (Figure 2) but in these beds the load of phosphorus removed (MRR) was higher than in the HF beds and amounted to 0.32 and 0.22 g/m²/day, respectively (Table 4). The effects of total phosphorus removal were much smaller in the VF type than in the HF type beds which was probably due to the shorter hydraulic retention time of wastewater in the VF type beds (Table 1).

The average efficiency of total phosphorus removal in the studied hybrid CWs was 90% and 94%, respectively (Figure 4). Similar phosphorus removal effects (94–96%) were obtained in hybrid CWs in Poland [20,41], Italy [23] and South Korea [75]. Much greater phosphorus removal effects (99%) were found in Denmark [17]. In most of the world's hybrid CWs the average efficiency of total phosphorus removal was usually between 70% and 89% [14,59,65,77,86]. There were also some cases where the efficiency of phosphorus removal was low and ranged from 47–65% [11,60,76] or even less than 40% [25,62,64]. It was also shown that the effects of total phosphorus removal in hybrid CWs are usually about 20–40% higher than in one-stage systems [20,38].

### 3.2. The Technological Reliability of the Studied Systems

The reliability of the wastewater treatment plant was determined using the Weibull method. In the first step, the parameters of the distribution were estimated and the null hypothesis that empirical data could be described by the Weibull distribution was verified. The datasets consisted of the concentration values of the main pollutants (TSS, BOD5, COD, total nitrogen, and total phosphorus) in the wastewater discharged from the HF bed. The null hypothesis was accepted. The results of the Hollander–Proschan goodness-of-fit test, along with the estimated parameters, are presented in Table 5.

**Table 5.** Parameters of the Weibull distribution and the Hollander–Proschan goodness-of-fit test.

| Parameter | Parameters of Weibull Distribution | | | Hollander–Proschan Goodness-of-Fit Test | |
|---|---|---|---|---|---|
| | $\theta$ | $c$ | $b$ | stat | $p$ |
| | Object No. 1—Zwierzyniec | | | | |
| TSS | −0.5000 | 1.8225 | 29.973 | −0.1096 | 0.9126 |
| BOD$_5$ | 1.0162 | 1.6296 | 4.0002 | 0.2118 | 0.8322 |
| COD | −1.0000 | 2.3924 | 39.104 | −0.0097 | 0.9922 |
| Total Nitrogen | 2.1111 | 2.0874 | 45.1160 | 0.0944 | 0.9247 |
| Total Phosphorus | 0.0000 | 1.2020 | 2.7683 | −0.2077 | 0.8354 |
| | Object No. 2—Florianka | | | | |
| TSS | 2.1061 | 1.5301 | 11.0160 | 0.1567 | 0.8754 |
| BOD$_5$ | −0.0500 | 1.7327 | 3.3194 | 0.1058 | 0.9157 |
| COD | −0.5000 | 2.3262 | 16.9940 | −0.3844 | 0.7006 |
| Total Nitrogen | 1.8636 | 1.0891 | 14.2960 | 0.2203 | 0.8255 |
| Total Phosphorus | −0.0500 | 2.1764 | 1.2472 | 0.8150 | 0.9350 |

Symbols: stat—value of the test statistic, $p$—significance level of the test; when $p \leq 0.05$ the data do not follow a Weibull distribution.

The goodness of fit of the obtained distributions at a significance level of $\alpha = 0.05$ was high: in the case of object No. 1 it ranged from 83% to 99% and in in the case of object No. 2 ranged from 70% to 93%.

The technological reliability of the wastewater treatment plant was determined on the basis of the cumulative distribution function (Figures 5–9), taking into account the limit concentration of the pollutants specified in the Polish Regulations [55] for wastewater treatment plants below 2000 PE.

The horizontal axis shows the concentration of pollutants in treated wastewater. The vertical axis shows the reliability percentage on a scale from 0 to 100. The graphs show the distribution with a 5% confidence interval. The graphs show the function dependence of the variable y on the independent variable *x*.

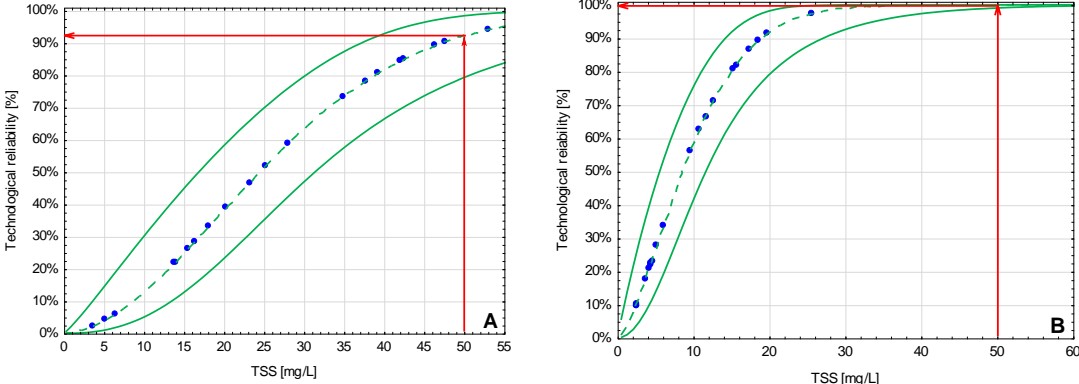

**Figure 5.** Weibull cumulative distribution functions and the technological reliabilities determined for TSS ((**A**)—object No. 1; (**B**)—object No. 2). Notation: dashed green line—reliability function, continuous green line—confidence intervals, red arrows—probability of achieving limit values of the indicators in the effluent.

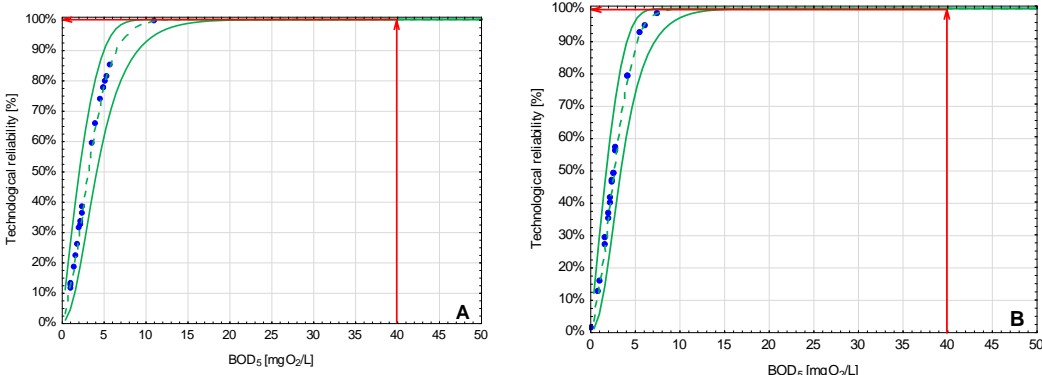

**Figure 6.** Weibull cumulative distribution functions and the technological reliabilities determined for BOD5 ((**A**)—object No. 1; (**B**)—object No. 2). Notation: dashed green line—reliability function, continuous green line—confidence intervals, red arrows—probability of achieving limit values of the indicators in the effluent.

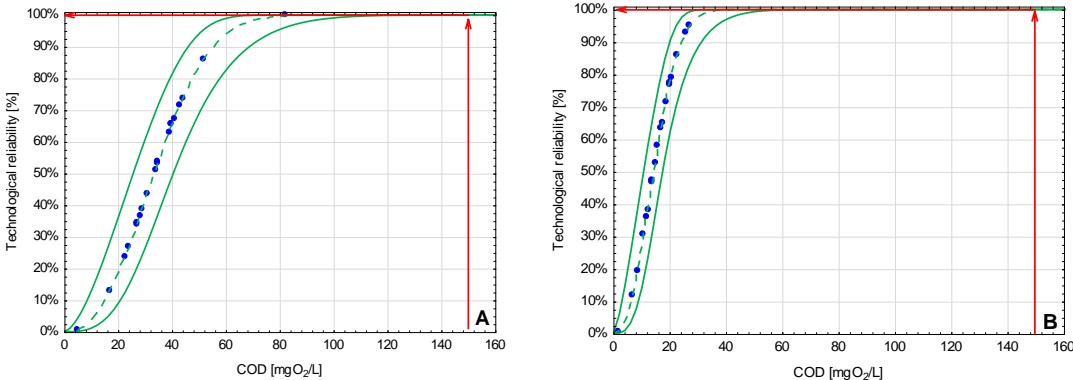

**Figure 7.** Weibull cumulative distribution functions and the technological reliabilities determined for COD ((**A**)—object No. 1; (**B**)—object No. 2). Notation: dashed green line—reliability function, continuous green line—confidence intervals, red arrows—probability of achieving limit values of the indicators in the effluent.

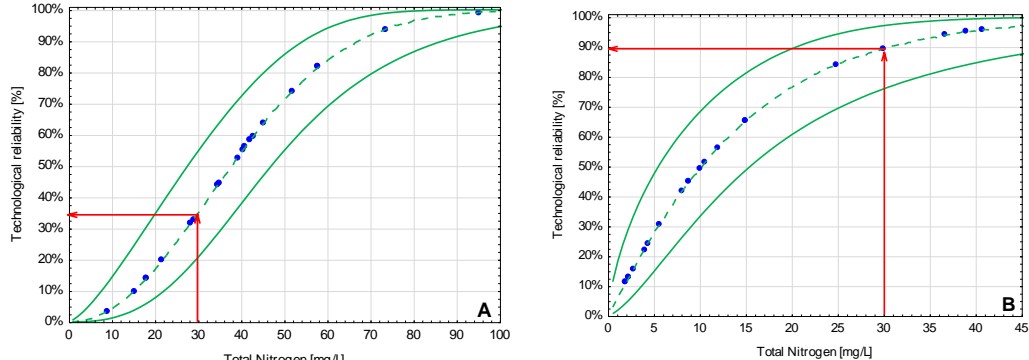

**Figure 8.** Weibull cumulative distribution functions and the technological reliabilities determined for total nitrogen ((**A**)—object No. 1; (**B**)—object No. 2). Notation: dashed green line—reliability function, continuous green line—confidence intervals, red arrows—probability of achieving limit values of the indicators in the effluent.

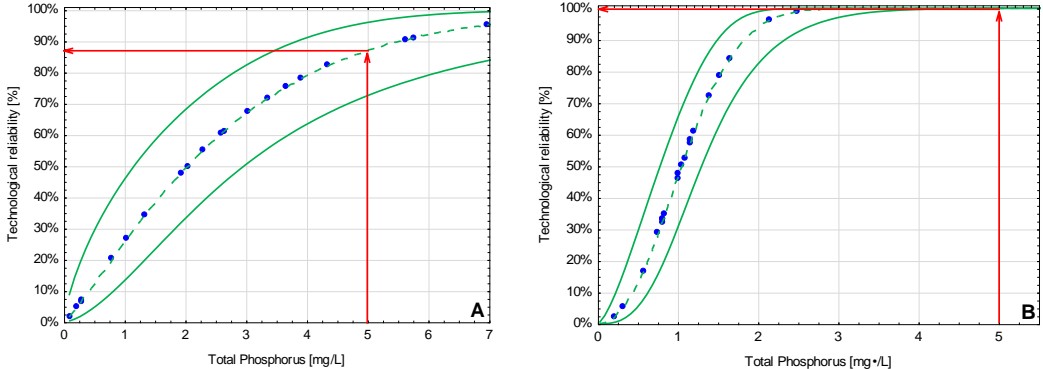

**Figure 9.** Weibull cumulative distribution functions and the technological reliabilities determined for total phosphorus ((**A**)—object No. 1; (**B**)—object No. 2). Notation: dashed green line—reliability function, continuous green line—confidence intervals, red arrows—probability of achieving limit values of the indicators in the effluent.

**TSS.** The reliability of removal of total suspended solids from wastewater in object No. 1 was 92% (Figure 5A). On this basis, it can be concluded that the plant operated smoothly on average 339 days a year. The period of failure-free operation is equivalent to the period when the concentration of total suspension particles in the wastewater discharged to the receiver was below the required limit (50 mg/L). According to the guidelines proposed by Andraka and Dzienis [87], a treatment plant characterised by a PE lower than 2000 should operate with a reliability of at least 97.3%, and a producer's risk of $\alpha = 0.05$: the treatment plant is allowed to be inoperable for a maximum of 9 days per year. When the number of days during which the TSS concentration in the treated wastewater exceeded the limit value was compared to the number of days in which the exceeded value had no negative impact on the assessment of plant operation, it was found that the concentrations of this parameter in the effluent were excessive, at 18 days a year.

The reliability of object No. 2 for total suspended solids removal was 100% (Figure 5B). On this basis, it can be concluded that with an operator's risk at the level of $\alpha = 0.05$, the values of this indicator in the treated wastewater throughout the whole year are lower than the permissible value, which is equivalent to failure-free operation of the facility. Very similar levels of total suspended solids removal reliability (93% and 100%) were found in the hybrid CWs inhabited by common reed, manna grass, and Virginia mallow [41], as well as giant miscanthus and Jerusalem artichoke [42]. In addition, TSS removal reliability at the level of 99.5% was achieved in a hybrid CW in a mountain eco-tourist farm in Poland [40]. A significantly lower reliability of TSS removal (90%) was obtained during long-term one-stage operation of CWs with *willow* [37].

**BOD5 and COD.** Both the assessment of the pollutant removal efficiency and the reliability analysis indicate that hybrid CWs in Poland provide very good conditions for the decomposition and removal of organic contaminants.

The reliability of removal of organic pollutants expressed by BOD5 and COD in both wastewater treatment plants was 100% (Figures 6 and 7). This means that the plants operated without any problems throughout the testing period, and the values of the tested parameters in the treated wastewater did not exceed the acceptable levels stipulated in the Polish law [55] (40 and 150 mg/L, respectively). This leads to the conclusion that, with an operator's risk at the level of $\alpha = 0.05$, the object should successfully pass inspection with regard to the parameters concerned throughout the year. The maximum (100%) reliability of BOD5 and COD removal in hybrid CWs operating under similar climatic conditions in Poland was also stated by Marzec et al. [41] and Jucherski et al. [40]. A lower reliability of BOD5 and COD removal (92% and 98%, respectively) was obtained during the long-term operation of a one-stage CWs with *willow* [37].

**Total nitrogen.** The probability that the concentration of total nitrogen in treated wastewater will not exceed the normative value, determined for wastewater discharged to standing waters from treatment plants of less than 2000 PE (30 mg/L) was 35% for object No. 1 and 89% for object No. 2 (Figure 8). The lower reliability of the total nitrogen removal in object No. 1 was related to the fact that the denitrification process in the HF-type bed was less intensive due to the coal deficit, which is necessary for this process. This also resulted in a lower efficiency of nitrate nitrogen, and, consequently, total nitrogen reduction (Figure 4).

The obtained reliability level indicates that for object No. 1 the permissible concentrations of total nitrogen in treated wastewater can be observed for 128 days per year, while for the rest of the year (237 days) the object did not meet the expected requirements. In turn, in the case of object No. 2, the concentration of total nitrogen exceeded the admissible value of 40 days a year. Taking into account the guidelines of Andraka and Dzienis [87], the total nitrogen concentration in treated wastewater, which exceeded normative values, may negatively affect the assessment of object No. 1 during 228 days of the year, while object No. 2 this may be the case for 31 days.

The obtained results indicate that in temperate climate conditions, with the limit values that are in force in Poland, the technological reliability of the removal of total nitrogen in hybrid CWs fluctuates greatly and is not always sufficient. Similar conclusions were reached by Marzec et al. [42]. They analysed the hybrid VF–HF type CWs with giant miscanthus and Jerusalem artichoke and then determined its reliability for nitrogen removal to be at the level of 32%. Additionally, in a one-stage CW during many years of operation, the reliability of nitrogen removal was only 45% [38]. A better level of reliability of total nitrogen removal (77%) was obtained by Jucherski et al. [40] in a hybrid CW in a mountain eco-tourist farm in Poland. Moreover, Marzec et al. [41], in a hybrid CWs with manna grass, common reed and Virginia mallow, achieved a 94% level of reliability of nitrogen removal.

**Total phosphorus.** The probability that the concentration of total phosphorus in treated wastewater flowing out of object No. 1 will reach the value below 5 mg/L was 87% (Figure 9). It was calculated that the acceptable value of this parameter was exceeded for 47 days a year. According to the guidelines of Andraka and Dzienis [87], excessive concentrations of total phosphorus negatively influence the evaluation of phosphorus removal reliability in object No. 1 for 38 days a year. On the other hand, the reliability of total phosphorus removal in object No. 2 was 100%. This means that with the operator's risk at the level of $\alpha = 0.05$, the concentration of total phosphorus in the treated wastewater throughout the year will not exceed the admissible level of 5 mg/L and the treatment plant would pass the control procedure with a positive assessment.

The relatively high reliability of total phosphorus removal may indicate that this indicator has less of a dependance on atmospheric conditions than is the case for total nitrogen. The results obtained are similar to those recorded by Jucherski et al. [40] and Marzec et al. [41], who in hybrid CWs obtained a phosphorus removal reliability at the level of 95% and 100%, respectively. A much lower reliability of phosphorus removal (28%) was found in hybrid VF–HF type CWs with giant miscanthus and Jerusalem

artichoke [42], as well as in a one-stage CW (48%) during long-term operation [41]. In another paper, Jóźwiakowski et al. [39] showed that the lowest reliability indexes for the pollutant removal were usually obtained in winter and spring, which may prove the influence of low air temperatures on the efficiency of wastewater treatment and reliability of operation in one-stage CWs.

## 4. Conclusions

The results of the research indicate a high efficiency of pollutant removal in both of the studied CWs. The efficiency of organic matter removal—expressed by $BOD_5$, COD, and TSS—and biogenic compound removal in the CWs in the Roztocze National Park was much higher than that obtained in one-stage CWs during long-term exploitation [37,38,88].

The analysed hybrid CWs provide significantly higher operational reliability compared to other technological solutions used for domestic wastewater treatment plants [89]. This is very important given the fact that they operate in the area covered by the highest form of nature protection and, therefore, they should meet the most stringent requirements.

The obtained results indicate that hybrid VF–HF type constructed wetlands should be recommended for use in protected areas for wastewater treatment and for water resource quality protection.

**Author Contributions:** Conceptualization, K.J. Data curation, K.J., M.M. Formal analysis, M.M., K.J. Investigation, A.M. and A.L. Methodology, K.J. Resources, A.M., K.J. Supervision, M.M., A.L. Validation, K.J. Writing—original draft, A.M. and K.J. Writing—review and editing, K.J., M.M. All authors have read and agreed to the published version of the manuscript.

**Funding:** This research received no external funding.

**Acknowledgments:** This paper was written on the basis of the research projects funded by: Polish Ministry of Science and Higher Education—project entitled: Water, wastewater and energy management (contracts No. TKD/DS1, TKD/S/1, 2017–2019), Provincial Fund for Environmental Protection and Water Management in Lublin, Poland—project entitled: Research on the optimization of the operation of household wastewater treatment plants at forest settlements in the area of Roztoczański and Poleski National Park (contract No. TKD/OŚ/37/2018), Roztoczański National Park, Poland—project entitled: Performing research on the operation of 6 household wastewater treatment plants located at forest settlements in the Roztoczański National Park (contract No. TKD/U-1/IŚGiE, 2019).

**Conflicts of Interest:** The authors declare no conflict of interest.

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
