# Peer review of "Technological Reliability and Efficiency of Wastewater Treatment in Two Hybrid Constructed Wetlands in the Roztocze National Park (Poland)"

_water, doi:10.3390/w12123435_

Round 1

Reviewer 1 Report

Introduction:

-It should be explained why Authors used reed and willow.

Abstract:

-Line 12. Please check the font type.

Results and Discussion

-Lines 216-220. This text should be rewritten. It is not clear for Readers. According the text, the TSS concentration was 71 mg/L in wastewater outflowing from the initial settling tank, and 129 mg/L in inflowing to the 1st VF type beds. However, in Tables 2 and 3, these values correspond to object no. 1 and the object no. 2.

-In my opinion, the composition of the wastewater flowing into the initial settling tank should be given in Table 3 or written as a text.

-On The Figures in section 3.1 the error bars should be given.

-General remark. Please do not only write the values of the pollutants concentration from the literature, but try to explain the difference between your results and the results reported in literature. Please indicate the difference in experiments, what is new in your study?, your study is better? Make the text more interesting for the Reader. Your results are very interesting and very important, but discussion is not well prepared. Please rewrite this section 3.1. Some examples are listed below.

-Could you explain why the TSS removal obtained on the VF type beds (Object No. 1) was lower compared to literature data? Please discussed this in the text.

-Line 260. Please explain what is the reason for such a wide range of TSS retention HF type CWs with different plants. The composition of the outflowing wastewater?

-Line 304. The same remarks. Please explained why the values (DO) obtained by you are lower than reported in literature.

-Lines 439-443. Please explain the difference in the PT removal in the VF and HF type beds?

Author Response

Response to Reviewer 1

Introduction:

-It should be explained why Authors used reed and willow.

We made the change according to Your suggestion.

The aim of this work is to present 3-year results of a study conducted on the technological reliability and the efficiency of domestic wastewater treatment in two hybrid CWs of VF-HF type with common reed (Phragmites australis (Cav.) Trin. ex Steud and willow (Salix viminalis L.) located on the area of the Roztocze National Park (Poland). In the analyzed objects reed and willow were used because previous 25-years of research performed in Poland have shown that these plants effectively support the processes of pollutants removal in CWs [46].

Abstract:

-Line 12. Please check the font type.

We made the change according to Your suggestion.

Results and Discussion

-Lines 216-220. This text should be rewritten. It is not clear for Readers. According the text, the TSS concentration was 71 mg/L in wastewater outflowing from the initial settling tank, and 129 mg/L in inflowing to the 1st VF type beds. However, in Tables 2 and 3, these values correspond to object no. 1 and the object no. 2.

We made the change according to Your suggestion.

The average values of pollutants in wastewater inflowing to the 1st VF type beds in the studied systems were presented in the Table 2 and 3. The received values were similar to the ones presented in the literature for mechanically treated wastewater in the initial settling tanks [37-38, 40-42].

-In my opinion, the composition of the wastewater flowing into the initial settling tank should be given in Table 3 or written as a text.

The composition of the wastewater flowing into the initial settling was described in an earlier paper [50]. Therefore, these results are not reported in this work.
Micek A.; JóĹşwiakowski K.; Marzec M.; Listosz A.; Malik A. Efficiency of pollution removal in preliminary settling tanks of household wastewater treatment plants in the Roztocze National Park. J. Ecol. Eng. 2020, 21 (5), 9–18.

-On The Figures in section 3.1 the error bars should be given.

We made the changes according to Your suggestion.

-General remark. Please do not only write the values of the pollutants concentration from the literature, but try to explain the difference between your results and the results reported in literature. Please indicate the difference in experiments, what is new in your study?, your study is better? Make the text more interesting for the Reader. Your results are very interesting and very important, but discussion is not well prepared. Please rewrite this section 3.1. Some examples are listed below.

-Could you explain why the TSS removal obtained on the VF type beds (Object No. 1) was lower compared to literature data? Please discussed this in the text.

We made the changes according to Your suggestion.

The efficiency of total suspended solids removal in the studied VF beds was lower than in similar objects of this type, because the wastewater inflowing to the beds contained a small concentration of TSS (71-129 mg/L) (Table 2, 3), as a large part of them (42-60%) were removed in 3-chamber settling tanks [54].

-Line 260. Please explain what is the reason for such a wide range of TSS retention HF type CWs with different plants. The composition of the outflowing wastewater?

We made the changes according to Your suggestion.

The diversified efficiency of TSS removal in different HF systems around the world is probably caused by the operation of these facilities in different climatic conditions and the use of different plants and materials to fill the bed as well as depths of the beds.

-Line 304. The same remarks. Please explained why the values (DO) obtained by you are lower than reported in literature.

We made the changes according to Your suggestion.

The obtained research results indicate that the oxygen concentration in the wastewater outflowing from the analyzed VF type beds were higher than previously reported in the literature [70-71], which proves that these beds provided better conditions for the decomposition of organic matter.

-Lines 439-443. Please explain the difference in the PT removal in the VF and HF type beds?

We made the changes according to Your suggestion.

The effect of total phosphorus removal were much smaller in VF type than in HF type beds which was probably due to the shorter hydraulic retention time of wastewater in the VF type beds (Table 1).

Reviewer 2 Report

General comments:

The authors observed removal efficiency and reliability of two hybrid constructed wetland systems treating domestic wastewater for 3 years. I suggest to make this manuscript more focused and concise, without unnecessary description citation and repeat. 

Specific comments:

-Abstract: What kind of domestic wastewater? How to address the reliability quantitatively?

L18: Change “96-99%” to “96%–99%”, the same below.

-Introduction:

The first two paragraphs are unnecessary. In addition to their locations, are there any innovative methods or findings in hybrid CWs field?

-Materials & Methods:

The first paragraph of “2.1. Characteristics of the Roztocze National Park” and Figure 1 are cited only, which could be deleted.

Figure 2: Not found in reference 40.

L159-160: Should be specify in details.

-Results & Discussion: Should be more concise. No need to repeatedly describle the values presented in figures/tables.

Figures 3-5, Table 4: Error bars or S.E. should be added.

-Conclusions: Should avoid repeating from methods and results. What are the major conclusions on the relationships? What are the clear implications?

L590: Specify which pollutants.

Author Response

Response to Reviewer 2

General comments:

The authors observed removal efficiency and reliability of two hybrid constructed wetland systems treating domestic wastewater for 3 years. I suggest to make this manuscript more focused and concise, without unnecessary description citation and repeat. 

Specific comments:

-Abstract: What kind of domestic wastewater?

We studied typical real domestic wastewater outflowing from single-family houses (from kitchen, bathroom, toilet).

How to address the reliability quantitatively?

Explanation: Technological reliability expressed in% is read from the distribution function and determines the probability of obtaining a value of the indicator lower than the permissible value in the tested sample, with the assumed risk (5%) that this condition will not be met (the value in the sample will exceed the normative value). In basic terms, reliability was interpreted in the work as the number of days in a year when the treatment plant operated properly, which is equivalent to not exceeding the limit value in treated sewage. On this basis, reliability can be considered purely quantitatively, as the amount of wastewater in which the value of a given indicator did not exceed the limit value in relation to the total amount of wastewater discharged into the environment in a given period. In this case, in order to calculate the amount of wastewater that meets the legal requirements, it is necessary to refer the level of reliability to the total amount of treated wastewater discharged from the facility (or it may be the number of days in a year when the sewage treatment plant operated correctly compared to the average daily wastewater discharge). Such an interpretation was not the subject of the work.

Part from the text of the work:

The Weibull distribution is a general probability distribution function used for reliability testing and failure risk assessment over time.  It found application, among others in device lifetime modelling and is flexible enough to replicate the key phases of the risk function run. With regard to wastewater treatment, the reliability analysis based on the Weibull distribution allows to determine the probability of occurrence of a certain value of pollution indicators in treated wastewater, and thus to assess the risk of exceeding the limit values. Thanks to it, it is possible to determine the time of defective operation of the treatment plant, which may be useful in making a decision regarding the operation of the treatment plant and its modernization [37-42].

L18: Change “96-99%” to “96%–99%”, the same below.

We made the changes according to Your suggestion.

-Introduction:

The first two paragraphs are unnecessary.

We confirm that the first paragraph is unnecessary and was delete. But the authors of the paper believe that the second paragraph should remain, as it is an introduction to the subject of work.

In addition to their locations, are there any innovative methods or findings in hybrid CWs field?

Yes. We have in Poland other hybrid constructed wetlands. They were described in our earlier paper: JóĹşwiakowski K., Marzec M., Kowalczyk-JuĹ›ko A., GiziĹ„ska-Górna M., Pytka-WoszczyĹ‚o A., Malik A., Listosz A., Gajewska M. 2019. 25 years of research and experiences about the application of constructed wetlands in southeastern Poland. Ecological Engineering 127, 440-45
But hybrid constructed wetlands presented in this paper are the first objects in Polish national park.

-Materials & Methods:

The first paragraph of “2.1. Characteristics of the Roztocze National Park” and Figure 1 are cited only, which could be deleted.

We made the changes according to Your suggestion.

Figure 2: Not found in reference 40.

It is this paper:

[50] JóĹşwiakowski K.; Marzec M.; GiziĹ„ska-Górna M.; Pytka A.; SkwarzyĹ„ska A.; Gajewska M.; SĹ‚owik T.; Kowalczyk-JuĹ›ko A.; Steszuk A.; Grabowski T.; Szawara Z. The concept of construction of hybrid constructed wetland for wastewater treatment in RoztoczaĹ„ski National Park. Bar. Reg. 2014, 12 (4), 91–102.

L159-160: Should be specify in details.

During the research period, 20 series of analyses were performed, during which 60 wastewater samples from 3 points of each object were taken and analysed (figure 1).

-Results & Discussion: Should be more concise. No need to repeatedly describle the values presented in figures/tables.

We made the changes according to Your suggestion.

Figures 3-5, Table 4: Error bars or S.E. should be added.

We made the changes according to Your suggestion

-Conclusions: Should avoid repeating from methods and results. What are the major conclusions on the relationships? What are the clear implications?

We made the changes according to Your suggestion

L590: Specify which pollutants.

L590 was deleted according to the previous recommendation

Reviewer 3 Report

This manuscript discusses the technological reliability and the efficiency of domestic wastewater treatment in two hybrid CWs of VF-HF type in the Roztocze National Park (Poland). The data collection (2017-2019) of this study includes 20 series of analyses where wastewater samples were collected. Based on my revision, the content is interesting for publication and needs few improvements before accepting it. The following minor comments are indicated as follows:

General comments:

  1. The topology of the Roztocze National Park is necessary to be clearly observed.
  2. What are the types of the Laboratory tests that conducted for Water and Wastewater in this study?
  3. The advantages and limitations of the Weibull method are suggested to be indicated.
  4. According to this study, the data from 2017 to 2019 have been collected and 60 wastewater samples were collected. It is necessary to observe; Have all data been considered in the period mentioned in this work and is there any noisy data was omitted from this study or not?
  5. It is suggested to add a discussion section to indicate how this work can be benefit for the reader.

Title:

I recommend the authors to summarize the title again for representing the main content in a simple way.

Specific Comments:

  • Abstract is very long, need to shorten into 150-200 words;
  • Scientific writing needs to be improved, some of the paragraphs has only one or two sentences, which makes the logic flow become poor, e.g. Line 30-34, 35-38.
  • Line 11-14: it will be interesting if you start the abstract with the problem statement that let you to do this work. Otherwise, I recommend you to summarize this abstract to be 150 -200 words.
  • Line 12-17: I suggest you to clearly indicate the utilized methodology in this work.
  • Line 45-48: authors indicated that it is necessary to build water supply and wastewater treatment systems that should not interfere with natural environment. Therefore, I recommend you to indicate and discuss the recent publication in this field of water quality in famous international journals, which can support your work, e.g. (https://doi.org/10.1016/j.jclepro.2020.120758; https://doi.org/10.1016/j.scitotenv.2020.141618.)
  • Line 70-78: What is the difference between the CWs in (one-stage and in the hybrid stage)?
  • The factors affect water supply and wastewater treatment are necessary to be considered and discussed (e.g. https://doi.org/10.1016/1016/j.jclepro.2020.124542; https://doi.org/10.1016/j.watres.2020.116437).
  • Based on figure 1, the scale and north direction are suggested to be indicated.
  • Line 119-120: what is the difference between the three hybrid CWs?
  • Line 141: based on Table 1, authors displayed some technological parameters of the studied objects. Why the authors selected these specific parameters, what are the advantages of these parameters than others?
  • Line 269: The photos in Fig. need to be numbered and discussed in more details
  • Line 291-240: I suggest you to summarize this section as much as possible.
  • Line 470-480: what s the difference between Fig. 6a and Fig. 6b?
  • Line 580-581: what is the similarity between this study and the previous works of Jucherski et al. [37] and Marzec et al. [38]?

Author Response

Reviewer 3

This manuscript discusses the technological reliability and the efficiency of domestic wastewater treatment in two hybrid CWs of VF-HF type in the Roztocze National Park (Poland). The data collection (2017-2019) of this study includes 20 series of analyses where wastewater samples were collected. Based on my revision, the content is interesting for publication and needs few improvements before accepting it. The following minor comments are indicated as follows:

General comments:

  1. The topology of the Roztocze National Park is necessary to be clearly observed.

Following the recommendation of the 2nd reviewer, the first part of description of the Roztocze National Park and Figure 1 have been removed. Therefore, the topology of the Roztocze National Park will not be specified in this paper.

  1. What are the types of the Laboratory tests that conducted for Water and Wastewater in this study?

The following parameters were determined: total suspended solids (TSS), BOD5, COD, total nitrogen (TN), total phosphorus (TP), pH, dissolved oxygen (DO), nitrate nitrogen, nitrite nitrogen and ammonium nitrogen.

  1. The advantages and limitations of the Weibull method are suggested to be indicated.

The description of the technological reliability analysis shows the advantages and possibilities of using the Weibull distribution. Limitations in the use of the method are related to the values of the parameters of the distribution of test data used in the analysis and their adjustment to the Weibull distribution. They were presented in chapter 2.4. and 3.2.

  1. According to this study, the data from 2017 to 2019 have been collected and 60 wastewater samples were collected. It is necessary to observe; Have all data been considered in the period mentioned in this work and is there any noisy data was omitted from this study or not?

All data been considered in the period mentioned in this work. No data was omitted from the analysis.

  1. It is suggested to add a discussion section to indicate how this work can be benefit for the reader.

The obtained results were discussed with literature from the world together with the description of the results in chapter "3. Results and discussion". Designating an additional chapter for discussion would be very difficult and would significantly extend the work.

Title:

I recommend the authors to summarize the title again for representing the main content in a simple way.

We made the change the title according to Your suggestion.

Technological reliability and efficiency of wastewater treatment in two hybrid constructed wetlands in the Roztocze National Park (Poland)

Specific Comments:

  • Abstract is very long, need to shorten into 150-200 words;

The abstract has been shortened according to Your suggestion – 203 words.

  • Scientific writing needs to be improved, some of the paragraphs has only one or two sentences, which makes the logic flow become poor, e.g. Line 30-34, 35-38.

According to reviewer 2 line 30-34 were deleted.

We made the change of line 35-38 according to Your suggestion:

National parks are institutions of great socio-educational importance, therefore their activities should be linked to the process of education about sustainable development and environmental protection [1]. In such protected areas not only nature but also air, soil and water protection is necessary [2-3].

  • Line 11-14: it will be interesting if you start the abstract with the problem statement that let you to do this work. Otherwise, I recommend you to summarize this abstract to be 150 -200 words.

The abstract has been shortened according to Your suggestion – 203 words.

  • Line 12-17: I suggest you to clearly indicate the utilized methodology in this work.

We inserted the methodology to the abstract according to Your suggestion.

Based on the obtained results the effects of pollutant removal and the technological reliability were determined, which was specified with the Weibull method.

  • Line 45-48: authors indicated that it is necessary to build water supply and wastewater treatment systems that should not interfere with natural environment. Therefore, I recommend you to indicate and discuss the recent publication in this field of water quality in famous international journals, which can support your work, e.g. (https://doi.org/10.1016/j.jclepro.2020.120758; https://doi.org/10.1016/j.scitotenv.2020.141618.)

We made the change according to Your suggestion. We cited the indicated papers.

Properly implemented water and wastewater management has a significant impact on limiting the eutrophication process of surface waters and reduces the degradation of groundwater quality [5-8].

  • Line 70-78: What is the difference between the CWs in (one-stage and in the hybrid stage)?

One-stage CWs have one bed and hybrid systems have two or three beds.

  • The factors affect water supply and wastewater treatment are necessary to be considered and discussed (e.g. https://doi.org/10.1016/1016/j.jclepro.2020.124542 ; https://doi.org/10.1016/j.watres.2020.116437).

We made the change according to Your suggestion. We cited the indicated papers

Properly implemented water and wastewater management has a significant impact on limiting the eutrophication process of surface waters and reduces the degradation of groundwater quality [5-8].

  • Based on figure 1, the scale and north direction are suggested to be indicated.

Following the recommendation of the 2nd reviewer Figure 1 have been removed.

  • Line 119-120: what is the difference between the three hybrid CWs?

These objects have different amount of inflowing wastewater as well as beds surfaces, but they use the same plants, i.e. reed and willow.

  • Line 141: based on Table 1, authors displayed some technological parameters of the studied objects. Why the authors selected these specific parameters, what are the advantages of these parameters than others?

Table 1 presents the most important technological parameters of the tested wastewater treatment plants, which indicate their size as well as their components and construction.

  • Line 269: The photos in Fig. need to be numbered and discussed in more details

We made the changes according to Your suggestion. Figure 5 has been corrected and descriptions of the results have been added to the text.

  • Line 291-240: I suggest you to summarize this section as much as possible.

Summarize of lines 240-291 is rather impossible, because this fragment presents the research results and contains a discussion of the obtained results with the world literature.

  • Line 470-480: what s the difference between Fig. 6a and Fig. 6b?

Figure 6 presents Weibull cumulative distribution functions and the technological reliabilities determined for TSS. Figure A – for object No. 1, Figure B – for object No. 2. The differences in the reliability values are described in the text.

  • Line 580-581: what is the similarity between this study and the previous works of Jucherski et al. [37] and Marzec et al. [38]?

The obtained research results and their comparison with the literature, including the papers of Jucherski et al. [37] and Marzec et al. [38] are presented in Chapter 3. Also the second reviewer recommended that "Conclusions: Should avoid repeating from methods and results".

Round 2

Reviewer 1 Report

Many thanks to the Authors for their comprehensive responses to my comments. The paper is ready to be published in the journal.

Author Response

Thank you very much for your valuable comments and acceptance of our paper.

Reviewer 2 Report

As the authors' response, what the CW treated was typical real domestic wastewater outflowing from single-family houses (from kitchen, bathroom, toilet). That’s common in CW field. Although the authors’ revisions are technically accepted, one or more highlights in CW field should be refined and presented. Location in National Park is not really an innovation. I recommend it to be accepted after minor revision.

Author Response

Thank you very much for your valuable comments and acceptance of our paper.

We have corrected Highlights.

Reviewer 3 Report

no comment. Suggest to accept.

Author Response

(The authors gave the same response as above.)
